# Lady’s Bedstraw as a Powerful Antioxidant for Attenuation of Doxorubicin-Induced Cardiotoxicity

**DOI:** 10.3390/antiox12061277

**Published:** 2023-06-15

**Authors:** Jovana Bradic, Marijana Andjic, Jovana Novakovic, Aleksandar Kocovic, Marina Tomovic, Anica Petrovic, Marina Nikolic, Slobodanka Mitrovic, Vladimir Jakovljevic, Danijela Pecarski

**Affiliations:** 1Department of Pharmacy, Faculty of Medical Sciences, University of Kragujevac, 34000 Kragujevac, Serbia; jovana.jeremic@medf.kg.ac.rs (J.N.); salekkg91@gmail.com (A.K.); marinapop@gmail.com (M.T.); petkovicanica0@gmail.com (A.P.); 2Center of Excellence for Redox Balance Research in Cardiovascular and Metabolic Disorders, Svetozara Makovica 69, 34000 Kragujevac, Serbia; marina.rankovic.95@gmail.com (M.N.); drvladakgbg@yahoo.com (V.J.); 3Department of Physiology, Faculty of Medical Sciences, University of Kragujevac, 34000 Kragujevac, Serbia; 4Department of Pathology, Faculty of Medical Sciences, University of Kragujevac, 34000 Kragujevac, Serbia; smitrovic@medf.kg.ac.rs; 5Department of Human Pathology, 1st Moscow State Medical, University IM Sechenov, Moscow 119991, Russia; 6The College of Health Science, Academy of Applied Studies Belgrade, 11000 Belgrade, Serbia; danijela.pecarski@assb.edu.rs

**Keywords:** lady’s bedstraw, doxorubicin-induced cardiotoxicity, oxidative stress, cardiac function, rat

## Abstract

This study aimed to examine the effects of a 14-day treatment with lady’s bedstraw methanol extract on doxorubicin-induced cardiotoxicity through functional, biochemical and histological examinations. We used 24 male *Wistar albino* rats divided into the following groups: control (CTRL), doxorubicin (DOX), and DOX + GVE (*Galium verum* extract). GVE was administered orally at a dose of 50 mg/kg per day for 14 days, while a single dose of doxorubicin was injected into the DOX groups. After accomplishing treatment with GVE, cardiac function was assessed, which determined the redox state. During the autoregulation protocol on the Langendorff apparatus, ex vivo cardiodynamic parameters were measured. Our results demonstrated that the consumption of GVE effectively suppressed the disturbed response of the heart to changes in perfusion pressures caused by administration of DOX. Intake of GVE was associated with a reduction in most of the measured prooxidants in comparison to the DOX group. Moreover, this extract was capable of increasing the activity of the antioxidant defense system. Morphometric analyses showed that rat hearts treated with DOX showed more pronounced degenerative changes and necrosis compared to the CTRL group. However, GVE pretreatment seems to be able to prevent the pathological injuries caused by DOX injection via decrease in oxidative stress and apoptosis.

## 1. Introduction

Doxorubicin represents an important cytotoxic chemotherapy agent that has been widely used in the treatment of breast, lung, thyroid cancers, as well as other solid and hematologic cancers. The mechanism of anti-tumoral activity is based on the intercalation into DNA base pairs, and the inhibition of topoisomerase II (TOP2) in cancer cells resulting in DNA damage and cell death [1]. Nevertheless, one of the main concerns regarding doxorubicin administration that strongly limits its use involves cardiotoxicity. Reports suggest that cardiotoxicity induced by anthracycline derivatives accounts for more than 30% of cardiotoxicity related to chemotherapy, thus requiring a specific approach for prevention and alleviation of cardiac adverse effects [1]. Initially, doxorubicin-induced heart damage reflects in asymptomatic dysfunction, while it progresses to serious cardiomyopathy and heart failure. Reactive oxygen species (ROS) production and accumulation has been highlighted as one of the main contributors of doxorubicin-induced cardiotoxicity, along with disrupted Ca^2+^ homeostasis and mitochondrial function [2]. In that sense, treatment protocols, including the administration of antioxidants, might be a crucial step in the alleviation of oxidative stress and triggering cardioprotection.

Lady’s bedstraw (*Galium verum*, *G. verum*), which belongs to the Rubiaceae family, has been used in traditional medicine due to its antioxidant, anticancer, anti-inflammatory, analgesic, antioxidant effects. In addition to the wide spectrum of known pharmacological effects encountered in the traditional use of the plant, the role of lady’s bedstraw in cancer prevention and treatment has also been documented by preclinical studies [3]. The antioxidant and anticancer potential of this plant species is dedicated to the presence of polyphenols, iridoid, glycoside, anthracene derivatives and other biologically active compounds [3,4,5]. Previously conducted research in our laboratory has confirmed the significant cardioprotective and antioxidant capacity of the methanol extract of *G. verum* in ex vivo models of ischemia/reperfusion injury in rats. To the best of our knowledge, there are no data regarding the impact of *G. verum* extract in DOX-induced cardiac damage. Taking into consideration the beneficial effects of this extract on the functional recovery of the heart after ischemia mediated via a decrease in oxidative stress, we hypothesized that *G. verum* would significantly contribute to the alleviation of doxorubicin-mediated injury in rat hearts.

Based on the aforementioned data, the aim of our study was to assess the influence of a 14-day application of *G. verum* methanol extract on cardiac dysfunction induced by DOX in rats. The novelty of our research reflects an understanding of the influence of *G. verum* extract on cardiotoxicity induced by DOX, which has been completely unknown so far, by performing functional, biochemical, and structural analyses of heart.

## 2. Materials and Methods

### 2.1. Plant Material and Extract Preparation

The aerial parts of *G. verum* were collected in July, 2021 in the village Dobroselica, on the southern cliff of the mountain Zlatibor (GPS coordinates: 43°42′59.99″ N and 19°41′59.99″ E). The collected material was dried and ground to pass through a 0.75 sieve. The extract was prepared by heat reflux extraction, a temperature of 900 °C, for a duration of 2 h using methanol as a solvent [6]. A total of 100 g of the aerial part of the plant was extracted with 500 mL of methanol, and the extraction yield was 12.3 g per 100 g of the used plant. Afterwards, a rotary evaporator was used to obtain the dry extract, which was dissolved in tap water daily, before being applied to animals, in order to mimic the oral intake in the human population in accordance with the previous literature data [4].

### 2.2. Analysis of Individual Phenolic Compounds (LC-DAD-MS/MS Analysis)

The methanol extract was analyzed by HPLC with LC-MS/MS detection in order to characterize the phytochemical content. Qualitative and quantitative analyses of 26 major constituents were carried out [7]. The extract was first dissolved in DMSO in concentrations of 2%, 0.2%, and 0.02%, after which it was filtered through a membrane filter made of regenerated cellulose with a pore size of 0.45 µm. The HPLC method used had been previously well-established for qualitative and quantitative analysis. The sample size was 5 μL, and separation was performed on Agilent Technologies 1200 HPLC-DAD coupled to Agilent Technologies 6410A ESI-QqQ-MS/MS (LC-DAD-MS/MS), and Zorbax Eclipse XDB-C18 column (50 mm length, 4.6 mm width, 1.8 μm particle size) thermostated at 50 °C was used. The mobile phase contained two components: (A) A 0.05% aqueous solution of formic acid; and (B) Methanol. The flow rate of the mobile phase was 1.0 mL/min, and the following gradient mode was used: 0–6 min 30% Component B, 6–9 min 70% Component B, and 9–12 min 100% Component B. A UV/VIS signal in the range 190–700 nm was monitored for the purposes of possible identity confirmation. The effluent was passed to the MS/MS detector without splitting the flow. The ion source parameters were: nebulizer pressure 40 psi, drying gas temperature and flow (N2) 350 °C and 9 L/min, capillary voltage 4000 V, and negative polarity.

### 2.3. Determination of the Total Phenolic and Flavonoid Contents and Evaluation of Antioxidant Activity

The Folin–Ciocalteu’s method was used to determine the total phenolic content (TPC) of *G. verum* extract [8]. TPC was expressed as mg gallic acid equivalents per gram of dry extract (mg GA/g DE). The total flavonoid content (TFC) of the tested extract was determined based on flavonoid affinity to form a complex with aluminum chloride (AlCl3). The mean of three readings and TFC expressed as milligrams of quercetin equivalents per gram of dry extract (mg QE/g of DE) were used [9].

The investigated *G. verum* extract was tested using a DPPH (1,1-diphenyl-2-picrylhydrazyl) assay, using a spectrophotometric method adapted for microplates [10,11,12]. Synthetic antioxidant BHT served as a positive control. The radical-scavenging capacity (RSC) was calculated in accordance with the previously published research.

### 2.4. Ethics Statement

This investigation was conducted in the laboratory for cardiovascular physiology of the Faculty of Medical Sciences, University of Kragujevac, Serbia. The experimental protocol was approved by the Ethics Committee of the Faculty of Medical Sciences, University of Kragujevac, Kragujevac, Serbia, number: 01-2772. All experiments involving animals were conducted in strict accordance with the European Directive for Protection of the Vertebrate Animals used for Experimental (86/609/EEC) and the principles of Good Laboratory Practice (GLP).

### 2.5. Experimental Animals

The twenty-four healthy male *Wistar albino* rats (eight weeks old, body-weight 300 ± 30 g) included in this study were obtained from the Military Medical Academy, Belgrade, Serbia. The animals were housed under controlled regular environmental conditions, at room temperature (22 ± 2 °C), with 12 h of automatic illumination daily. The animals consumed commercial rat food and tap water ad libitum. After a one-week environment adaptation, the rats were divided into three equal groups as follows:CTRL—healthy untreated rats;DOX—rats treated with a single DOX injection;GVE + DOX—rats treated with *G. verum* extract dissolved in tap water and single DOX injection.

### 2.6. Drug Administration

The rats in the CTRL group were healthy untreated rats that consumed commercial rat food and tap water ad libitum for two weeks. On the other hand, the animals in the DOX group were rats that consumed commercial rat food and tap water ad libitum, while the rats forming the GVE + DOX group consumed commercial food and GVE dissolved in tap water for the 2 weeks prior to the DOX treatment. To mimic the most common and the easiest administration route, *G. verum* extract was applied *per os*, and the rats were kept in separate cages to provide a precise amount of drug for each animal. The individual extract dose adjusted to the rat’s body weight (50 mg/kg) was dissolved in 50 mL of water, considering that this volume corresponds to the daily water consumption in rats. Treatment with *G. verum* extract was repeated every day, once daily, for the duration of two weeks. [13].

After the two-week protocol, the rats from both the DOX and the GVE + DOX groups received an acute intraperitoneal injection of DOX at a dose of 15 mg/kg body weight (DOX was dissolved in dimethyl sulfoxide–DMSO, the volume of DMSO ranged from 0.45 to 0.6 mL) in order to develop cardiomyopathy. Moreover, the rats from the CTRL group received an injection with the same volume of DMSO [13]. The rats from all three groups were sacrificed 72 h after the DMSO or DOX injection [14]. The animals were anesthetized using an intraperitoneal injection including a mixture of ketamine (10 mg/kg) and xylazine (5 mg/kg) and sacrificed by decapitation.

### 2.7. Assessment of Ex Vivo Cardiac Function

After the sacrifice and emergency thoracotomy, the heart was quickly isolated, immersed in cold saline, and retrogradely perfused through the aorta in a Langendorf apparatus with Krebs–Henseleit solution. The composition of the Krebs–Henseleit solution was as follows (mmol/L): NaCl 118, KCl 4.7, CaCl_2_ × 2H_2_O 2.5, MgSO_4_ × 7H_2_O 1.7, NaHCO_3_ 25, KH_2_PO_4_ 1.2, glucose 11, pyruvate 2, equilibrated with 95% O_2_ plus 5% CO_2_, and warmed to 37 °C (pH 7.4).

After the incised left atrium, the mitral valves were separated and a transducer was inserted into the left ventricle to measure the heart performance. The following parameters of the myocardial function were continuously measured (at each perfusion pressure): the maximum and minimum rates of pressure development in the left ventricle (dp/dt max, dp/dt min), systolic and diastolic left ventricular pressure (SLVP, DLVP), and heart rate (HR). Coronary flow (CF) was measured flowmetrically.

In order to examine the heart function, after the stabilization period, the perfusion pressure was gradually decreased to 60 cm H_2_O, and then increased to 80, 100 and 120 cm H_2_O (pressure changing protocol 1, PCP 1). Subsequently, the perfusion pressure was decreased to 40 cm H_2_O, and the same protocol was repeated, increasing the pressure from 40 to 120 cm H_2_O (pressure changing protocol 2, PCP 2) [14].

After accomplishing ex vivo experiments, the hearts were frontally excised with a scalpel into two halves and further processed depending upon the analysis of parameters. One part was immediately fixed in 4% neutral formalin for histopathological analysis, while the second part was immediately frozen at −80° for the estimation of the cardiac oxidative stress.

### 2.8. Heart Tissue Preparation

The parts of hearts that were used to determine oxidative stress parameters in cardiac tissue were immediately frozen at −80 °C. The 0.5 g section of each heart tissue were homogenized in 5 mL of ice-cold phosphate-buffered saline (pH 7.4) using an electrical homogenizer. Then, the homogenate was centrifuged at 1200× *g* at 4 °C for 20 min. The supernatants were separated and stored at −80 °C until the biochemical analyses were performed. The index of lipid peroxidation was measured as a thiobarbituric acid reactive substance (TBARS), as well as a parameter of the antioxidant defense system; reduced glutathione (GSH), catalase (CAT), and superoxide dismutase (SOD), were determined in heart tissue [15].

### 2.9. Systemic Redox State

The blood samples were collected from the jugular vein in the moment of the animals’ sacrifice to evaluate the systemic redox state. The heparinized venous blood samples were centrifuged in order to separate the plasma and erythrocytes, which were stored at −20 °C until biochemical analysis. The following pro-oxidant parameters were determined from plasma samples: superoxide hydrogen peroxide (H_2_O_2_), anion radical (O_2_^−^), nitrites (NO_2_^−^), and TBARS. The parameters of the antioxidative defense system were determined from the erythrocyte lysate samples, the activity of SOD and CAT and the level of GSH, as previously described [16].

### 2.10. Histological Analysis

The heart samples were fixed in 4% neutral formalin for at least 24 h for histological examination. After fixation, tissue sections were dehydrated, embedded in paraffin under a vacuum, and sectioned at 5 µm thickness on a rotating microtome (Leica, Wetzlar, Germany). The 5 μm thin sections were stained with the standard hematoxylin and eosin-H&E. The degree of the damage was assessed according to the following previously established scoring method: no change = 0 score; focal changes = 1 score (focal areas of myonecrosis, edema, and inflammation); patchy changes = 2 score (patchy areas of myonecrosis, edema, and inflammation); confluent changes = 3 score (confluent areas of myonecrosis, edema, and inflammation; massive changes = 4 score (massive areas of myonecrosis, edema, and inflammation) [17]. Additionally, 5 μm thick sections of cardiac tissue were dewaxed, rehydrated, and treated with citrate buffer (pH 6.0) in the microwave oven for antigen detection in order to perform immunohistochemical staining. Staining was visualized by using the EXPOSE Rabbit specific HRP/DAB detection IHC Kit (ab80437, Abcam, Cambridge, UK), and sections were counterstained with Mayer’s hematoxylin. The slices were also incubated with cardiac troponin T-cTnT (ab209813), recombinant anti-Bax (ab32503), Bcl2 (ab32124), anti-cleaved Caspase 3 (ab2302), and anti-heat shock protein 70 (HSP70; ab5439) overnight at room temperature. Sections were photomicrographed with a digital camera mounted on a light microscope (Olympus Corporation, BX51, Shinjuku-ku, Tokyo, Japan), digitized and analyzed [13]. These analyses were performed by a pathology specialist.

### 2.11. Statistical Analysis

IBM SPSS 20.0 for Windows was used for statistical data processing. The Shapiro–Wilk test was used to examine the normality of the distribution. Data are expressed as mean value ± standard deviation and analyzed by one-way analysis of variance (ANOVA) tests, followed by a Tukey’s post hoc test for multiple comparisons when the distribution between groups was normal. Kruskal–Wallis was used for comparison between groups where the distribution of data was different than normal. A value of *p* < 0.05 was considered statistically significant.

## 3. Results

### 3.1. LC-DAD-MS/MS Analysis of GV Extracts

The chemical composition of the *G. verum* methanol extract determined by LC-DAD-MS/MS analysis is shown in Table 1. The findings suggested the presence of various bioactive molecules, while the most abundant compounds were chlorogenic acid, cynaroside, and ursolic acid (Figure 1).

### 3.2. Total Phenolic and Total Flavonoid Content

The total phenolic and flavonoid contents of *G. verum* extract are represented in Table 2.

### 3.3. Antioxidant Activity

The antioxidant activity of *G. verum* extract examined by performing DPPH free radical scavenging assays is shown in Table 3.

### 3.4. Ex Vivo Cardiac Function

The values of ex vivo-measured cardiac function parameters and coronary flow, during the pressure changing protocols on the Langendorff apparatus are presented in Figure 2. To examine the potential difference due to *G. verum* extract treatment, we compared the percentage of decrease (−) or increase (+) between PCP 1 and PCP 2 in all the groups (Table 4).

The most obvious differences in the cardiodynamic parameters, as well as the coronary flow, between PCP 1 and PCP 2 were observed in the DOX group. Major changes in parameters of heart contractility were noticed at 60 cm H_2_O for dp/dt max (−28.55), as well as for dp/dt min (−31.15). These parameters did not significantly change during pressure changing in a group of animals treated with *G. verum* extract followed by single DOX administration.

The values of the left ventricular pressures indicate evident differences during changing pressure in the DOX group. The large differences between pressure protocols observed for SLVP were noticed at 80 cm (−35.77), 100 cm (−43.88), and 120 cm H_2_O (−42.93). On the other hand, the marked change values of DLPV were at 80 cm (−20.71) and 100 cm H_2_O (−22.73).

In addition, DOX treatment significantly changed HR during PCPs at 80 cm H_2_O (−21.46), as well as CF at 80 cm (−24.32), 100 cm (−23.78), and 120 cm H_2_O (−22.02).

### 3.5. Systemic Redox State

The level of pro-oxidative markers measured in the examined groups indicated that rats treated only with DOX administration were associated with the highest release of all oxidative stress parameters (Figure 3). The concentration of O_2_^−^, NO_2_^−^ and the level of TBARS were significantly higher in the DOX group compared to the CTRL and GVE + DOX groups. The two-week administration of *G. verum* extract caused a significant decrease in the mentioned parameters so that values in the GVE + DOX group were similar to the values in the CTRL group and decreased in comparison to the DOX group. The results of the measured antioxidant parameters are represented in Figure 3. The treatment with *G. verum* extract caused a significant elevation in the catalase activity in comparison to the DOX group, which was also significantly reduced in relation to the CTRL group. Additionally, the SOD activity has increased in the GVE + DOX group compared to the DOX group. On the other hand, the GSH activity did not change under the impact of GVE and DOX.

### 3.6. Oxidative Stress in Cardiac Tissue

Following a single DOX injection, the value of the pro-oxidative marker TBARS in the heart tissue was significantly higher in groups treated with DOX in comparison to the CTRL group. As a result of the DOX administration, the antioxidant defense system was weakened. SOD activity and the level of GSH were reduced in both DOX groups compared to the CTRL group. On the other hand, it is obvious that these parameters are higher in rats treated with GVE followed by DOX injection compared to DOX administration alone (Figure 4).

### 3.7. Histopathological Analysis

Photographs of H&E-stained sections are presented in Figure 5, while the degree of heart damage in all groups according to scoring method is presented in Table 5. In the H&E-stained coronary heart sections of rats in the CTRL group, there are discrete histological lesions that vary from hyperemia and edema to degenerative changes and unicellular or partially focal necrosis of muscle cells. Degenerative changes and necrosis are significantly more pronounced in the myocardium of animals in the DOX and DOX + GVE groups. The necrosis, in the DOX group, is mostly confluent and massive; there are zonal necroses in a larger number of muscle cells, with fiber fragmentation, loss of nuclei, and hypereosinophilia of the cytoplasm. On the other hand, in the DOX + GVE group, patchy and confluent-type fields are observed more often.

Additionally, these ultrastructural changes were further confirmed by the degree of immunohistochemical expression of cTnT. As shown in Figure 6, the highest expression of cTnT was noticed in the CTRL group. On the other hand, the lowest expression of cTnT with its highest depletion was noticed in the DOX group. Additionally, lower cTnT SOD depletion was revealed in the DOX + GVE group compared to the DOX group.

To investigate the ability of GVE to prevent DOX-induced apoptosis, we measured the expression of proapoptotic markers such as Caspase-3 and Bax, as well as antiapoptotic marker Bcl2 (Figure 7). The immunohistochemical analysis showed that administering of DOX resulted in the greatest Bax and Caspase-3 expression, which was significantly lower in DOX + GVE and discrete in the CTRL group. On the other hand, the highest expression of Bcl2 was noticed in the CTRL group; discrete expression was observed in DOX + GVE, while single-cell-focal expression was scarce in rats treated only with DOX.

Immunohistochemical analysis showed that Hsp70 expression was the highest in rats treated using a single intraperitoneal DOX injection (DOX group). Additionally, significantly lower expression of Hsp70 was revealed in the CTRL and group of rats treated with GVE (Figure 8).

## 4. Discussion

The clinical use of DOX, as one of the most effective agents prescribed against various tumors, is limited due to its serious cardiac side effects. Evidence strongly recommends applying dietary antioxidants, especially of natural origin, as a worthwhile tool in the preservation of heart function and structure after DOX therapy [2,18]. The wide distribution of *G. verum*, its use in traditional medicine, and its confirmed antioxidant effects make it favorable for examination as an adjuvant in the management of DOX-induced cardiotoxic effects. In the current study, we used a DOX-induced cardiotoxicity animal model to examine the effects of *G. verum* extract on cardiac function and structure, and for the first time, we provided data regarding the protective role of this plant species in the reduction in DOX-induced toxic effects on the heart. Specifically, the main finding of our study is that *G. verum* extract was able to minimize DOX-induced cardiac dysfunction in rats by decreasing oxidative stress and apoptosis. Importantly, a 14-day administration of this herb extract alleviated DOX-induced pathohistological abnormalities in heart tissue.

In the first part of our research, we aimed to perform a chemical analysis of the obtained extract to identify the major components. LC-DAD-MS/MS analysis suggested the presence of various bioactive molecules, while the most abundant compounds in methanol extract were chlorogenic acid, cynaroside, ursolic acid, quercetin, and isoquercetin. Our results are in concordance to previously conducted investigations [5]. In order to determine the antioxidant properties of the lady’s bedstraw extract, we performed a DPPH assay that confirmed that methanol extract activity was higher in relation to the used standard. Earlier research also proved that *G. verum* extract or its ingredients exert valuable antioxidant activity [19,20].

The impact of *G. verum* extract on DOX-induced cardiotoxicity was assessed through functional, biochemical and histological examinations in the rat model. The application of a single cumulative dose of DOX 15 mg/kg i.p. was sufficient to induce cardiomyopathy in rats, as has been previously established [21]. During an ex vivo retrograde perfusion on the Langendorff apparatus, a decrease in cardiac parameters was noticed in the DOX group. The DOX application contributed to the disturbance of cardiac contractility, which was reflected in significant changes in dp/dt max and dp/dt min values during pressure changes, especially in the normoxic conditions on the CPP = 60 cm H_2_O. These findings suggest that the heart had an altered response to the changes in perfusion pressures and the impaired contractile force of the myocardium. Additionally, the lusitropic property of the heart was also diminished, as manifested by the drop in dp/dt min values, as well as the systolic and diastolic heart capacities. The abovementioned results are in line with the literature data referring to the doxorubicin-induced functional dysfunction [22,23]. The investigations pointed to a few mechanisms for cardiac dysfunction provoked by DOX, such as increased production of ROS leading to oxidative stress damage, disrupted Ca^2+^ homeostasis, and impaired genes’ expression involved in the apoptosis and necrosis of cardiomyocates [24,25]. The group of rats treated with *G. verum* extract had similar values in almost all measured parameters of cardiac function, which is an indication of normal heart function. Values of dp/dt max and dp/dt min remained in the physiological range, which is of crucial significance for the myocardium to contract, and relax properly. Unchanged cardiac function parameters, such as SLVP and DLVP, may indicate that *G. verum* has a protective effect on systolic and diastolic heart function. Additionally, coronary flow had relatively similar values in the *G. verum*-treated group during pressure changes, thus suggesting the influence of this plant species on the preservation of the coronary circulation. Our results on the cardiodynamic functional heart parameters proved that *G. verum* methanol extract triggers cardioprotection, which is in accordance with previous research performed in our laboratory, but on the I/R injury model [15]. In addition, previous investigations showed that some plant extracts also reduced DOX-induced cardiotoxity by mainly improving ventricular function, ejection fraction, and systolic and diastolic pressures [22,26]. Those beneficial effects on the cardiovascular system are mainly due to the properties of polyphenol compounds such as chlorogenic acid, cynaroside, ursolic acid, quercetin and isoquercetin. Previous research suggested that extracts containing similar antioxidants as our extract, such as quercetin, chlorogenic acid, and isoquercetin, reduced DOX-induced toxicity by reducing oxidative stress and apoptosis by upregulating the VEGF-B/Akt/GSK-3β signaling pathway [22]. A recent study showed that improved heart rate in mice, in the DOX-provoked cardiotoxity model, is a consequence of orally administered *Dillenia pentagyna* hydroalcoholic extract, whose active constituents are similar to those in our extract [27]. Furthermore, some investigators proved that the novel formulation containing encapsulated phenols (resveratrol and quercetin) recovered heart function through a reduction in oxidative stress and by diminishing the activity of caspase 3/7 [28].

In the second part of our study, we aimed to assess the impact of a two-week *G. verum* extract application on markers of oxidative stress in blood and heart tissue samples. The special contribution of this work is in elucidating the impact of extract consumption on both the systemic production of pro-oxidants, as well as their local production in heart tissue. Moreover, we aimed to reveal if this extract might alter antioxidant levels to understand if antioxidant activity is achieved by the direct neutralization of free radicals and/or by enhancing antioxidant defense system capacity. Our results suggest that all pro-oxidants detected in plasma samples were significantly elevated in the DOX group in comparison to CTRL, thus confirming increased ROS generations as a crucial factor in the pathogenesis of DOX-induced heart injury [22]. Intake of *G. verum* extract was associated with reduction in most of the measured pro-oxidants (NO_2_^−^, O_2_^−,^ and TBARS levels) in comparison to the DOX group. Moreover, this extract was capable of increasing SOD and CAT activities, as opposed to the DOX group, thus revealing that the mechanism responsible for the decrease in oxidative stress is a combination of direct free radical scavenging activity of polyphenols and the elevation of antioxidant defense enzymes. The results on the systemic redox state correlate with the heart redox state, confirming that elevated ROS generation is undoubtedly present in the DOX group, and represent target molecules for the novel tools used for cardioprotection. The significant alleviation of oxidative stress noticed in the *G. verum* group highlights this plant as a promising candidate in saving a large number of patients receiving DOX. The decreased pro-oxidants level and the increased activity of antioxidant enzymes after administering of *G. verum* extract corroborated the previously confirmed antioxidant properties of this plant [19,20]. Compiling data from experimental studies indicates that plants rich in polyphenols have profound cardioprotective effects exerted by enhancing antioxidant defense system capacity [27], as well as improving mitochondrial function and endoplasmic reticulus stress via MFN2/PERK pathway activation [29]. Furthermore, it was reported that polyphenols protect the heart from oxidative stress by inhibiting nicotinamide adenine dinucleotide phosphate-oxidase (NADPH) oxidase and xanthine oxidase, as well as chelation of iron ions [30].

In order to induce the effects of *G. verum* extract on structural heart damage induced by DOX, we performed histological analysis of heart tissue. While the discrete histological lesions were observed in the CTRL group, rats treated with DOX alone exhibited notable myocardial damage, such as zonal necrosis, of a larger number of muscle cells, with fiber fragmentation, loss of nuclei, and hypereosinophilia of the cytoplasm. Cardiotoxicity in rats belonging to the DOX group was verified by significantly more pronounced degenerative changes and necrosis of muscle cells, with the highest score of myocardial damage compared to both CTRL and DOX + GVE. These findings correlate with the available evidence that strongly suggest DOX-induced myocardial damage [31,32]. On the other hand, DOX-induced cardiac alterations were less pronounced in rats treated with GVE, which had larger patchy and confluent areas. Literature data strongly suggest apoptosis as one of the main factors contributing to the loss of functional myocytes after DOX treatment, which leads to irreversible heart injury. To gain insight into the mechanisms included in the cardioprotective effects of *G. verum* extract, the expression of proapoptotic markers was determined in the heart tissue. Our results indicate increased expression of Bax and Caspase-3 and decreased Bcl-2 expression in DOX group, thus confirming that this anticancer agent induced apoptosis in myocardial tissue. These findings are consistent with previously conducted studies that revealed elevated expression of apoptotic markers and downregulation of antiapoptotic proteins after DOX exposure [33,34]. It has been proposed that oxidative stress can lead to Bax translocation to the outer mitochondrial membrane, which leads to the activation of caspase 3-dependent apoptosis in the heart [33]. On the other hand, we confirmed the alleviation of apoptosis in the hearts of rats treated with *G. verum* extract, as evidenced by the markedly lower expression of Bax and Caspase-3. In addition, phytochemical analysis of *G. verum* extract proved the presence of phenols and flavonoids, which we suppose, are responsible for the modification of the expression of Bcl-2, Bax and Caspase-3, as has previously been confirmed [35,36].

Additionally, severities of cardiac damage were assessed immunohistochemically by the determination of cTnT in cardiac tissue samples. The most prominent cardiac damage in the DOX group was evidenced by the lowest expression of cTnT in myocardial tissue. In brief, during myocardial damage, troponin T is released from the cardiomyocytes, while the loss of cTnT staining in necrotic areas indicates cardiac damage in the DOX group [13,37,38]. It is important to emphasize that higher expression and lower depletion of cTnT were noticed in group treated with *G. verum* extracts, thus suggesting that this plant species was able to prevent DOX-induced cardiac injury. We also aimed to evaluate whether *G. verum* extract may alter the expression of Hsp70 as one of the important molecular chaperones that has been implicated in various cellular pathways [39]. An increase in the expression of Hsp70 was noticed in in heart tissue of DOX rats, which is consistent with previous findings and once again confirmed that Hsp70 may act as a damage-associated molecule in the pathogenesis of DOX-induced cardiotoxicity [13,40]. Nevertheless, the consumption of *G. verum* extract was associated with the decreased values of Hsp70, thus suggesting the potential of this herb extract to preserve Hsp70 upregulation in the heart. These observations are in correlation with the markers of oxidative stress, which is expected knowing that elevated production of pro-oxidants induces Hsp70 expression [39]. Strikingly, a 2-week intake of *G. verum* extract was able to diminish histological alterations and the related oxidative stress and apoptosis following DOX treatment.

One of the limitations of our study reflects a lack of investigation of the influence of *G. verum* extract on the effectiveness of DOX as an anticancer drug. While several studies revealed the benefits of antioxidant consumption in reducing DOX-induced side effects [41], other authors have been concerned about the use of antioxidants in cancer therapy due to their potential to weaken the effects of DOX on tumor cells, and this topic is still debated in the scientific community [42]. It is important to emphasize that our results provide basic information referring to the potential use of lady’s bedstraw in minimizing cardiotoxicity related to DOX chemotherapy; however, the investigation of the impact of our extract on DOX effectiveness is imperative to ensure the clinical efficacy of this extract. Studies revealing whether *G. verum* extract alters the antitumor efficacy of DOX, in terms of weakening or potentially enhancing activity, are certainly necessary before its implementation as adjuvant in oncological patients.

## 5. Conclusions

Our results highlight that methanol *G. verum* extract improves cardiac function and effectively suppresses disturbed response of the heart to changes in perfusion pressures caused by the administration of DOX. Additionally, the consumption of *G. verum* attenuated oxidative stress and prevented the pathological injuries caused by DOX injection by reduction in apoptosis. In light of these findings, our research opens the potential of clinical studies on *G. verum* extract as a promising nutrition agent when combined with chemotherapy in order to diminish DOX-induced myocardial injuries. However, since this is the first study to consider the effects of *G. verum* extract on the function and structure of the isolated rat heart in the model of DOX-induced cardiotoxicity, future investigations should reveal the exact mechanism of action and elucidate the pharmacodynamics and pharmacokinetics of *G. verum* extract.

## Figures and Tables

**Figure 1 antioxidants-12-01277-f001:**
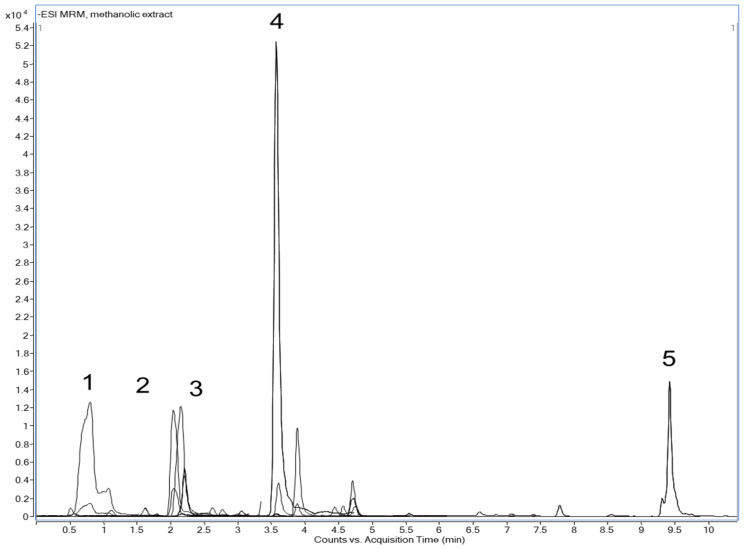
HPLC chromatogram of *G. Verum* methanol extract: compound 1—Chlorogenic acid; compound 2—Cynaroside; compound 3—Quercetin-3-O-glucoside; compound 4—Quercetin; compound 5—Luteolin.

**Figure 2 antioxidants-12-01277-f002:**
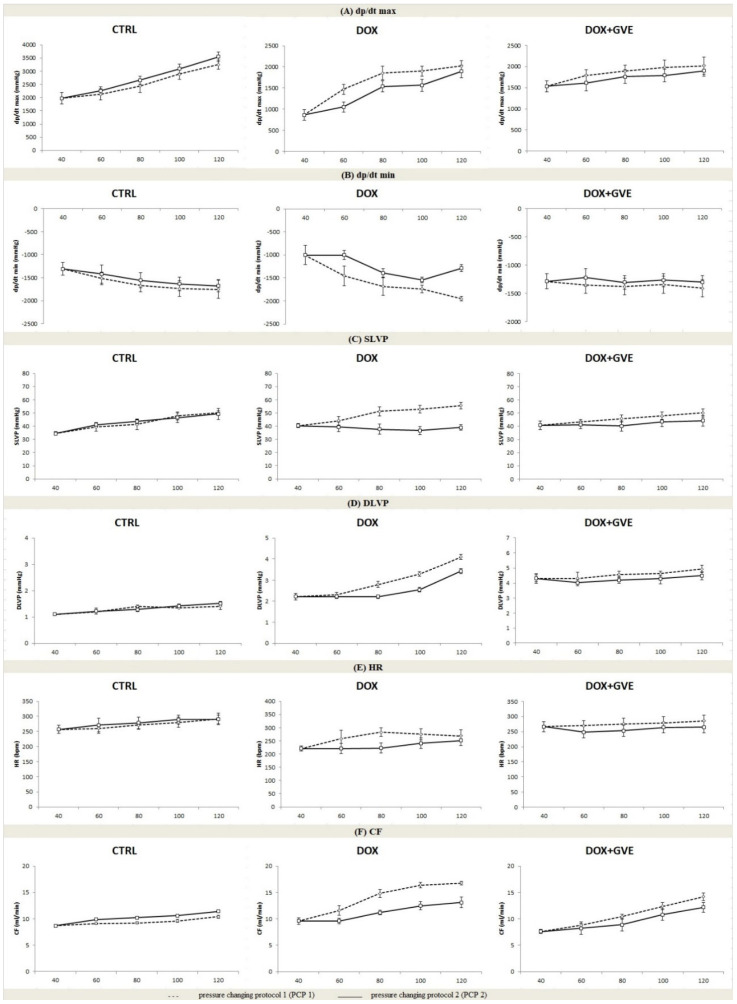
Effects of consumption of *G. verum* extract on ex vivo cardiac function parameters: (**A**) Values of dp/dt max within each of the 3 groups during PCP 1 and PCP 2; (**B**) Values of dp/dt min within each of the three groups during PCP 1 and PCP 2; (**C**) Values of SLVP within each of the three groups during PCP 1 and PCP 2; (**D**) Values of DLVP within each of the three groups during PCP 1 and PCP 2; (**E**) values of HR within each of the three groups during PCP 1 and PCP 2; (**F**) Values of CF within each of the three groups during PCP 1 and PCP 2. Values are expressed as mean ± standard deviation (n = 8). DOX—doxorubicin; GVE—*G. Verum* extract; CPP—constant perfusion pressure.

**Figure 3 antioxidants-12-01277-f003:**
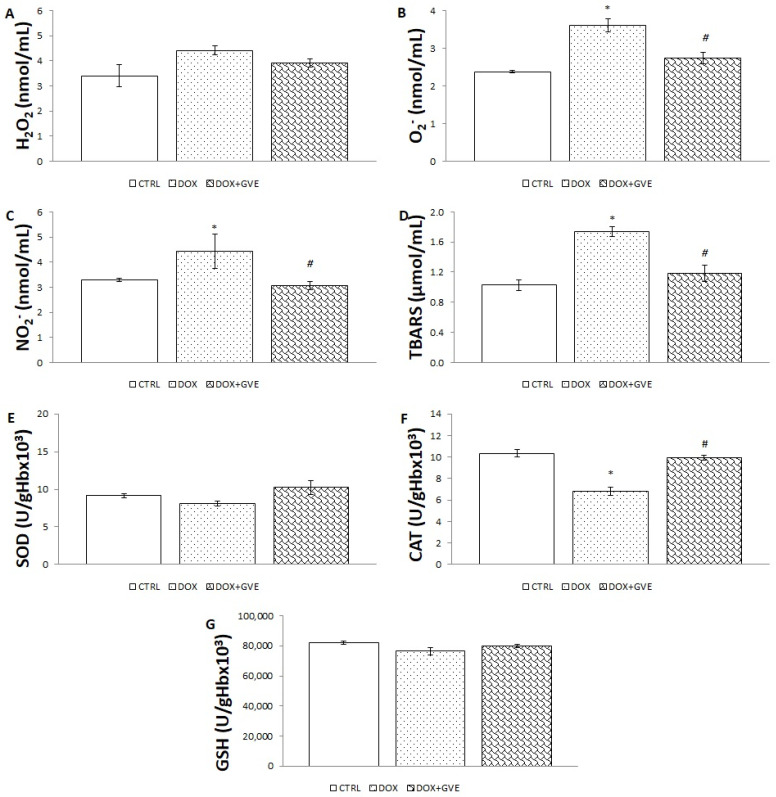
Effects of consumption of *G. verum* extract on the systemic oxidative stress parameters: (**A**) H_2_O_2_; (**B**) O_2_^−^; (**C**) NO_2_^−^; (**D**) TBARS; (**E**) SOD; (**F**) CAT; (**G**) GSH. Values are expressed as median. The upper bar shows the value of the third quartile and lower bar shows the first quartile (n = 8). * *p* < 0.05 compared to CTRL group; # *p* < 0.05 compared to DOX group.

**Figure 4 antioxidants-12-01277-f004:**
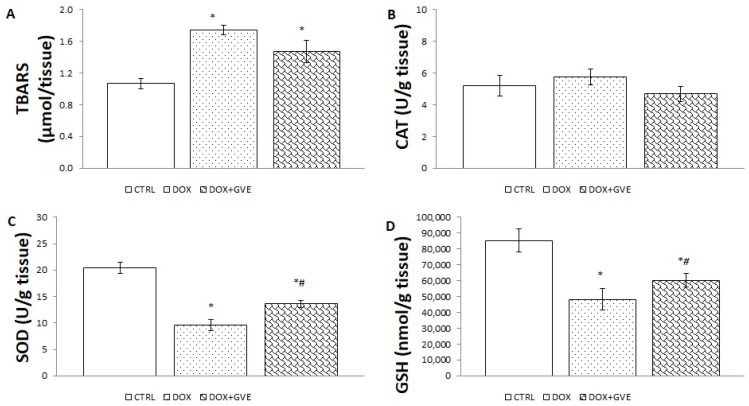
Effects of consumption of *G. verum* extract on the value of the antioxidant defense system in heart tissue: (**A**) TBARS; (**B**) CAT; (**C**) SOD; (**D**) GSH. Values are expressed as median. The upper bar shows the value of the third quartile and the lower bar shows the first quartile (n = 8). * *p* < 0.05 compared to CTRL group; # *p* < 0.05 compared to DOX group.

**Figure 5 antioxidants-12-01277-f005:**
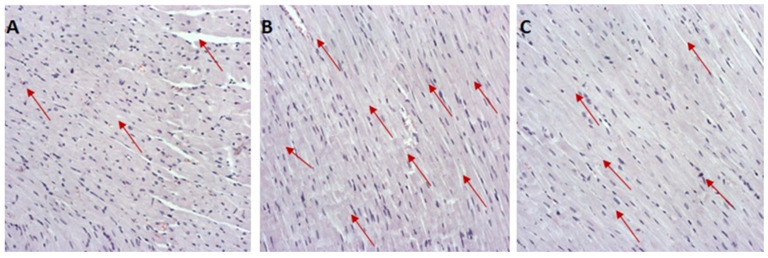
Representative heart tissue sections of hematoxylin/eosin-staining: (**A**) CTRL group; (**B**) DOX group; (**C**) DOX + GVE group. Red arrows indicate representative changes in the heart tissue. Original magnification 20×.

**Figure 6 antioxidants-12-01277-f006:**
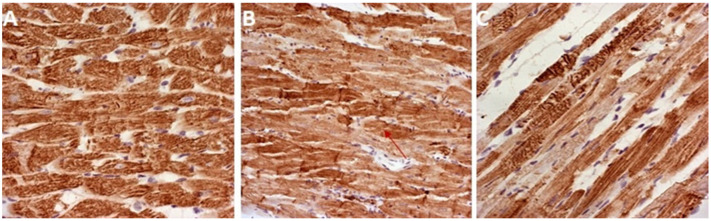
cTnT-immunostaining: (**A**) CTRL group; (**B**) DOX group; (**C**) DOX + GVE group. Original magnification 20×. Red arrow indicates the area with the least colored cTnT in the heart tissue.

**Figure 7 antioxidants-12-01277-f007:**
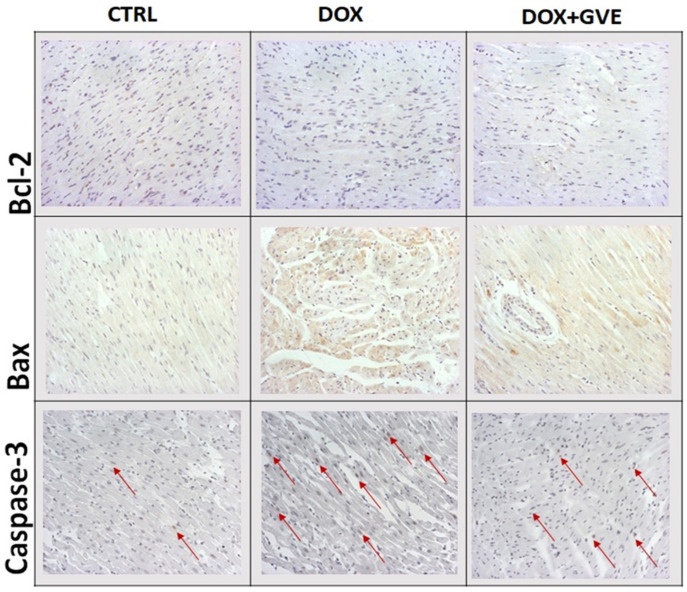
Representative heart tissue sections of Bcl-2, Bax and Caspase-3 staining in CTRL, DOX and DOX + GVE groups. Original magnification 20×.

**Figure 8 antioxidants-12-01277-f008:**
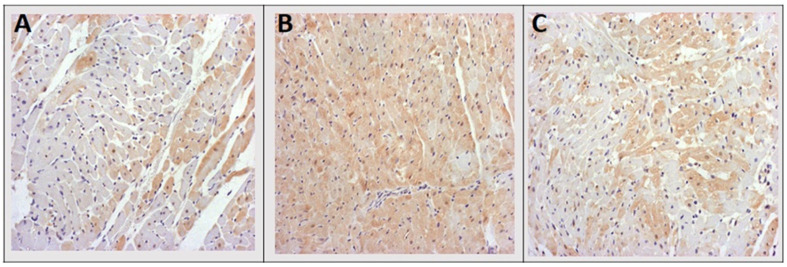
Representative heart tissue sections of immunohistochemical staining of Hsp 70. (**A**) CTRL group; (**B**) DOX group; (**C**) DOX + GVE group. Original magnification 20×.

**Table 1 antioxidants-12-01277-t001:** Quantitative and qualitative analysis of individual compounds found in aerial part of *G. verum* extract expressed as µg/g of dry extract.

Name of Compound	*G. verum* Extract	Retention Time [t_R_]	Name of Compound	*G. verum* Extract	Retention Time [t_R_]
Ursolic acid	8900 ± 66	9.5	p-Hydroxybenzoic acid	8.1 ± 0.8	1.08
Chlorogenic acid	15,562 ± 780	0.8	Kaempferol	7.5 ± 0.9	4.55
Isoquercetin	6871 ± 211	2.25	Hesperetin	14.2	4.44
Cynaroside	9611 ± 293	2.13	Rhamnetin	<4.79	5.84
Quercetin	179.2 ± 53.6	3.74	Diosmetin	12.3	5.02
p-Coumaric acid	54.2 ± 4.7	1.69	Rosmarinic acid	0.28	2.43
Astragalin	93.6 ± 3.6	2.8	Alizarin	<2.55	5.95
Caffeic acid	19.7 ± 1.2	1.18	Epicatechin	<0.60	0.95
Ferulic acid	17.6 ± 1.4	1.9	Glycyrrhizin	<0.14	7.41
Luteolin	33.8 ± 1.7	4.03	Hyperoside	<0.32	2.16
Isorhamnetin	60.1 ± 3.2	4.79	Catechin	<0.60	0.74
Apigetrin	45.3 ± 2.4	2.81	Myricetin	<19.51	2.67
Vanillic acid	14.5 ± 4.6	1.24	Ellagic acid	<0.30	2.61

**Table 2 antioxidants-12-01277-t002:** Total phenol and flavonoid contents of *G. verum* extract.

Extract	Total Phenolic Content (mg GAE/g DE)	Total Flavonoid Content (mg QE/g DE)
*G. verum* extract	27.58 ± 2.09	66.97 ± 4.17

Values are means of three biological replicates ± SD. GAE—gallic acid equivalent; QE—quercetin equivalent; DE—dry extract.

**Table 3 antioxidants-12-01277-t003:** DPPH free radical scavenging activity of *G. verum* extract.

DPPH Radical Scavenging Activity (IC_50_ ^a^ (µg/mL))		
Sample	BHT	*G. verum* Extract
	8.66 ± 0.12	11.78 ± 1.18

^a^ The mean value ± SE of three measurements; BHT—*tert*-butylated hydroxytoluene.

**Table 4 antioxidants-12-01277-t004:** Percentage differences between PCP 1 and PCP 2 during ex vivo perfusion.

CPP	CTRL	DOX	DOX + GVE	CPP	CTRL	DOX	DOX + GVE
dp/dt max				dp/dt min			
60	6,20	−28,55	−9,81	60	−5,08	−31,15	−8,13
80	7,24	−17,78	−7,55	80	−5,91	−17,28	−6,85
100	6,98	−13,71	−9,26	100	−5,31	−12,23	−8,34
120	6,23	−9,77	−6,98	120	−3,18	−26,03	−5,36
SLVP				DLVP			
60	4,57	−11,85	−5,03	60	1,67	−3,48	−6,46
80	5,33	−35,77	−12,23	80	−5,86	−20,71	−8,05
100	−2,93	−43,88	−9,41	100	3,97	−22,73	−7,34
120	−1,60	−42,93	−13,12	120	3,80	−16,10	−9,11
HR				CF			
60	4,27	−14,43	−7,92	60	8,79	−17,24	−6,82
80	2,58	−21,46	−8,20	80	10,87	−24,32	−13,62
100	3,21	−12,42	−5,44	100	10,42	−23,78	−12,90
120	−0,68	−6,41	−6,19	120	9,62	−22,02	−13,08

**Table 5 antioxidants-12-01277-t005:** Degree of heart damage.

Groups	Score
CTRL	0.8
DOX	3.5
DOX + GVE	2.5

No change = 0 score; focal changes = 1 score; patchy changes = 2 score; confluent changes = 3 score; massive changes = 4 score.

## Data Availability

Data is contained within the article.

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
