# Peer review of "Lady’s Bedstraw as a Powerful Antioxidant for Attenuation of Doxorubicin-Induced Cardiotoxicity"

_antioxidants, 2023, doi:10.3390/antiox12061277_

Round 1

Reviewer 1 Report

 This study aimed to examine the effects of 14-day treatment with Lady’s Bedstraw methanol 23 extract on doxorubicin-induced cardiotoxicity through functional, biochemical and histological examinations.

This is a well-written paper with little to comment on statistically. The investigators have controlled for the type 1 error within each outcome, but there are many outcomes. Do the investigators regard as different familywise comparison, and if not, then how are they controlling for the overall type 1 error? There are three arms in this study, but Figure 1 and tables 1-3 do not mention these. Is the information contained in these tables obtained over all 24 rats? In table 4, what is CPP and what does, for example, the values 40 and 40* signify? The results state, “These parameters did not significantly change during pressure 269 changing in a group of animals treated with G. verum extract followed by single DOX 270 administration.”  How do you know that this is not type 2 error, given your small sample size?

Reviewer 2 Report

Bradic et al explored the effect of Lady bedstraw on DOX cardiotoxicity in mice model. I have several comments:

Why did the authors choose 2 week time frame?

Why weren’t the controls treated with lady bedstraw as well, such group would provide very valuable information?

Could the authors provide data by which they chose 15 mg/kg dosage?

In consideration that n was 8, I highly doubt that distribution was normal (for most data), and thus it is not appropriate to use parametric tests (and also data should be presented as median and IQR accordingly)

Table 4. The most important results concerning cardiac function should be represented by figure in order to increase readability.

Figure 2 it seems to me that mean and SEM were shown rather than mean and SD as stated in the footnote.

Line 480 typo

Discussion should be substantially revised. The authors extensively comment their results but only briefly mention available studies, thus leaving the reader with no critical conclusions. In addition, clinical perspective should be added.

Minor edits are needed.

Round 2

Reviewer 2 Report

No further comments.